# Intrusion Detection in IoT Networks Using Deep Learning Algorithm

**Bambang Susilo** 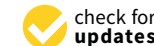 **and Riri Fitri Sari \***

Department of Electrical Engineering, Faculty of Engineering, Universitas Indonesia, Depok 16424, Indonesia; bambang.susilo91@ui.ac.id

\* Correspondence: riri@ui.ac.id

**Abstract:** The internet has become an inseparable part of human life, and the number of devices connected to the internet is increasing sharply. In particular, Internet of Things (IoT) devices have become a part of everyday human life. However, some challenges are increasing, and their solutions are not well defined. More and more challenges related to technology security concerning the IoT are arising. Many methods have been developed to secure IoT networks, but many more can still be developed. One proposed way to improve IoT security is to use machine learning. This research discusses several machine-learning and deep-learning strategies, as well as standard datasets for improving the security performance of the IoT. We developed an algorithm for detecting denial-of-service (DoS) attacks using a deep-learning algorithm. This research used the Python programming language with packages such as scikit-learn, Tensorflow, and Seaborn. We found that a deep-learning model could increase accuracy so that the mitigation of attacks that occur on an IoT network is as effective as possible.

**Keywords:** machine learning; deep learning; Internet of Things; distributed denial-of-service attack; intrusion detection

---

## 1. Introduction

The Internet of Things (IoT) is a very recent technology that interfaces devices through the internet, improving and supporting people's lives, careers, and cultures [1]. In 2017, Yuan et al. [2] suggested that 17 million denial-of-service attacks would happen by 2020. The IoT is one of the quickest developing online areas, with 50 billion connected gadgets expected before the end of 2020 [3].

Internet of Things frameworks are open around the world, essentially comprising compelled assets and developed by lossy connections. Accordingly, critical adjustments of existing security ideas for data and remote systems ought to be actualized to provide compelling IoT security techniques. When utilizing current security instruments, for instance, encryption, authentication, access control, network protection, and application control take time and are insufficient for a huge system among many associated machines, with every piece of the system having its own vulnerability. For instance, Mirai is an uncommon sort of botnet that triggers huge-scale distributed denial-of-service (DDoS) strikes by abusing IoT machines [4]. The Persirai thingbot is one variant of Mirai code that continues to grow and infect Internet Protocol (IP) cameras [5]. Current protection components ought to be improved to fit the IoT ecosystem [6]. Nonetheless, usage protection components are immediately defeated when facing against the predetermined protection risks with various type of attacks made by enemies to bypass current settings. As an example, an enhanced DDoS offense masquerades as the origin IP address in order to make the offensive area undetectable by safeguards. Subsequently, offensives that are more unpredictable and ruinous than Mirai have been observed as a common result of the vulnerabilities of IoT frameworks. Additionally, in this study, we sought to identify strategies in

IoT frameworks to face several vulnerabilities and scenarios of IoT applications [6]. Thus, creating successful IoT security strategies was a research focus in this work.

Many methods and frameworks to mitigate network attacks have been developed. Machine-learning and deep-learning methods, both supervised and unsupervised, can help to observe logs that can reach millions in a day on a large network. Furthermore, many researchers use network-attack datasets: the knowledge discovery and data (KDD) cup obtained from the International Knowledge Discovery and Data Mining Tools Competition, the NSL-KDD is dataset developed by the Canadian Institute for Cybersecurity at the University of New Brunswick, and the BoT-IoT is dataset developed by the University of New South Wales (UNSW) Canberra Cyber center.

In this research, the authors classify attacks by using an existing dataset. The classification of the attacks uses machine-learning (random forests: RF) and deep-learning methods (convolutional neural network: CNN; multi-layer perceptron: MLP). The results of the study are expected to be used in a network-based intrusion detection system (NIDS) to conduct anomaly detection on an IoT network.

This article is organized as follows. Section 2 introduces the security and deep-learning method. A machine-learning application in IoT security is presented in Section 3. The results and analysis appear in Section 4. Finally, in Section 5, we present the conclusion.

## 2. Security and Deep-Learning Method

### 2.1. IoT Protection

The IoT coordinates the internet and the real world to provide effective synergy between humans and the IoT environment. Normally, IoT appliances operate in various settings to achieve various objectives. Nonetheless, their activity must meet exhaustive security requirements in cyber and physical states. In this way, the security requirements with large-scale attack surfaces of the IoT framework need to be tested. To fulfill ideal security requirements, a comprehensive view of network security is needed [7–12].

Besides providing great benefits in various fields, such as those of the technical, economical, and social, there have been discussions of IoT risk using the MicroMort model and its effects on the economy, as well as a case study of the calculation of the MicroMort model with several Mirai code variants [5,13,14]. The following principal security properties ought to be considered while building up a compelling IoT security strategy [15,16].

Confidentiality: This is a fundamental protection norm for IoT frameworks. IoT appliances can save and move delicate data that do not to be illegally uncovered by people. Health check (patient privacy), individual, business, and armed forces information are exceptionally classified and need to be verified against illegal users [17].

Integrity: Information through IoT appliances is commonly moved via remote correspondence and needs to be uniquely altered with legitimate substances. Trustworthiness highlights are, in this manner, major in guaranteeing a compelling checking instrument to identify any adjustment during correspondence over an unreliable remote system. Respectability highlights can verify the IoT framework from pernicious information sources that may be utilized to dispatch structured query language (SQL) injection attacks.

Authentication: The characters of the element must be completed first before carrying out several other procedures. However, because of the concept of the IoT framework, there is a need for verification that differs from one framework to another. For instance, confirmation needs to be strong in the IoT framework wherever assistance is required to provide powerful protection instead of great adaptability. Trade-offs are a significant challenge in building up an authentication design, e.g., the trade-off between security and safety in IoT health appliances while planning an authentication design.

Authorization: This incorporates giving clients privileges for an IoT framework, such as a physical measuring appliance. Clients might use systems, people, or administration information, among other

kinds, that is gathered with computing devices and able to be approved by clients (e.g., official goods in warehouses and people who need information) [18].

Availability: The administration conveyed by IoT frameworks should consistently be accessible to authorized users. Accessibility is a central element of a fruitful arrangement of IoT frameworks. IoT frameworks and devices can now be rendered inaccessible by numerous dangers, such as DoS or active jamming. Accordingly, guaranteeing the persistent accessibility of IoT administration to clients is a fundamental aspect of IoT protection.

*2.2. IoT Threat*

Threats to security may be classified as virtual or real. Internet threats can also be active or passive [19–24]. The next paragraph provides a brief discussion of the risks.

2.2.1. Cyber Threats

Passive Threats: Latent risk is distinctly conducted via spying throughout correspondence feeds or the system. By listening in, the offensive can gather data from the instrument, the instrument owners, or both.

Active threats: The attack does not merely comprise the skillful listening of the correspondence routes but also the altering of IoT structures to modify their design, regulate their correspondence, refuse them any assistance, and so on. Attacks may incorporate a grouping of mediations, disturbances, and alterations.

Many kinds of DoS offensives could be used against an IoT. These could come from a regular offensive that is made to exhaust assets from a specialist co-op and re-arrange data-transfer capacity to mark that objective remote correspondence [25,26]. Distributed denial-of-service attacks are a serious offensive when there are few attacks propelled via multiple IPs, which create a separation from typical traffic [27–33].

2.2.2. Threats to the Physical

These dangers can be as bad as the demolition of the device. The offensive, for the most part, is not equipped with specialized abilities to direct an online attack. Hence, an attack just affect the accessible device items from different segments of the devices that are used for administration. By embracing IoT frameworks, these kinds of offenses can be widescale because the vast majority of the device objects (sensors and cameras) are open access [34]. These hazards might be caused by accidental harm from natural or human disasters like earthquakes, floods, or wars.

## 3. Method of Machine-Learning Application in IoT Security

*3.1. Dataset Description*

There are many types of datasets that have been used by researchers to test systems. One of the datasets utilized for this study was that developed by the UNSW, Canberra, who have several datasets; the one that we used in this experiment was a dataset called the BoT-IoT. This was made by constructing a surrounding system at the Cyber Range Laboratory in the UNSW Canberra Cyber center. The environment was made by combining normal traffic and botnets. Data sources were provided in various formats, including original file with a .pcap extension and comma separated values (CSV) files. Pcap files are generally used by Wireshark programs that contain data packets on a network. In general, this file is used for analyzing data characteristics on the network. The files are separated, based on attack categories and subcategories, to further assist in the labeling process [35]. Table 1 shows a summary of the number of attacks contained in the dataset.

**Table 1.** BoT-IoT (Internet of Things) attack summary of a dataset [35].

| Categorized | Attack | Port | Tools | Number |
|---|---|---|---|---|
| Information gathering | Service scanning | | Nmap, hping3 | 1,463,364 |
| | OS Finger Printing | | Nmap, xprobe2 | 358,275 |
| Denial of service | DDoS | TCP | hping3 | 19,547,603 |
| | | UDP | hping3 | 18,965,106 |
| | | HTTP | golden-eye | 19,771 |
| | DoS | TCP | hping3 | 12,315,997 |
| | | UDP | hping3 | 20,659,491 |
| | | HTTP | golden-eye | 29,706 |
| Information theft | Keylogging | | Metasploit | 1469 |
| | Data theft | | Metasploit | 118 |
| Total | | | | 73,360,900 |

OS: operating system; DDoS: distributed denial-of-service; DoS: denial-of-service; TCP: Transmission Control Protocol; UDP: User Datagram Protocol; HTTP: HyperText Transfer Protocol.

### 3.2. Machine-Learning and Deep-Learning Anomaly Detection

The experiment was done using a Dell notebook type G7 7590 with the Windows 10 Home 64-bit operating system. The processor was an Intel Core i7-9750H with a six-central processing unit (CPU) core. The notebook was equipped with 16 GB random access memory (RAM). The graphics processing unit (GPU) of the notebook was the Nvidia RTX 2060, with 6 GB RAM. Dell is an American multinational computer technology company, manufacturer in China, and sell in Indonesia. To perform data cleaning and feature selection, we used the Panda framework and the NumPy framework. After the process was run, the Matplotlib framework was used to display images. We also used the scikit-learn and Keras framework for data analysis, and we used Tensorflow to activate the GPU [36–39].

#### 3.2.1. Random Forests

Random forests is an algorithm used in the classification of large amounts of data. Classification is done by merging trees or several decision-tree algorithms by training the available sample data. The most selected class is chosen as the final classification output [7].

#### 3.2.2. Support Vector Machine (SVM)

Support-vector machines (SVMs) are usually used for classification (such as support-vector classification) and regression (support-vector regression). In classification modeling, an SVM has a more mature and more mathematically clear concept compared to other classification techniques. Support-vector machines can also overcome the problem of classification and regression in a linear or non-linear way. Therefore, statistical learning is a baseline form of an SVM [7].

#### 3.2.3. Multilayer Perceptron

Multilayer perceptron is a feedforward neural network that has a number of neurons or nerves that are connected with other neurons with connecting weight neurons, where every neuron that exists is a unit that has the task of processing and calculating the activation value, which symbolizes the set of predecessors of each unit from input to output or from one unit to another unit.

#### 3.2.4. Convolutional Neural Network

CNNs has the tasked with reducing the information parameters utilized in an artificial neural network (ANN). A CNN comprises an input and an output layer, as well as multiple hidden layers. The information factors are decreased by using equivariant representation, parameter sharing, and three sparse interactions. Decreasing associations between the layers enhances the training time difficulty and extends the scalability of a CNN [7].

### 3.3. Evaluation Criteria

The modeling of objective functions was used to determine the outcome of the RF, MLP, and CNN algorithms. The scenario used in the study was set up for the work (performance) of each machine-learning technique tested on the dataset used. Measures (metrics) that can be used in assessing this performance include accuracy, true positive rate, and true negative rate [40–43].

A performance matrix comes from a confusion matrix. A confusion matrix is a table that visualizes the performance of a classification algorithm tested versus actual classification.

Accuracy: This is the degree of proximity between the predicted value and the actual value. Accuracy is a way of assessing categorization models. Equation (1) depicts a single-class accuracy measurement:

$$Accuracy = \frac{TP + TN}{TP + TN + FP + FN}. \tag{1}$$

Precision: This describes the level of accuracy between the requested data and the predicted results provided by the model. Thus, precision is the ratio of true positive predictions compared to overall positive predicted results. Precision values can be obtained by the Equation (2):

$$Precision = \frac{TP}{TP + FP}, \tag{2}$$

where *TP* is true positive, *TN* is true negative, *FP* is false positive, and *FN* is false negative.

Table 2 shows the parameters we used for the CNN and MLP. For the activation function, we used rectified linear unit (ReLU) and softmax for both classifier algorithms. We also utilized the Adam optimizer in the experiment.

**Table 2.** Parameters for deep-learning algorithms.

| Algorithm | Batch Size | Activation Function | Optimizer | Epochs |
|---|---|---|---|---|
| Convolutional Neural Network (CNN) | 32, 64, 128 | ReLU and softmax | Adam | 10, 30, 50 |
| Multilayer Perceptron (MLP) | 32, 64, 128 | ReLU and softmax | Adam | 10, 30, 50 |

Figure 1 depicts the training and testing algorithm of the process of the dataset.

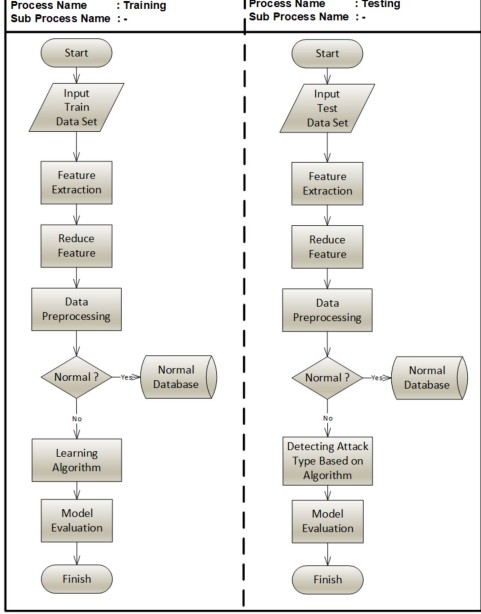

**Figure 1.** Flowchart training and testing algorithm.

Figure 2 can be divided into three main parts:

a.     Embedding blocks, to apply context.
b.     Convolution and max-pooling blocks, for extracting dataset features.
c.     Dense and activation blocks, for learning and classification.

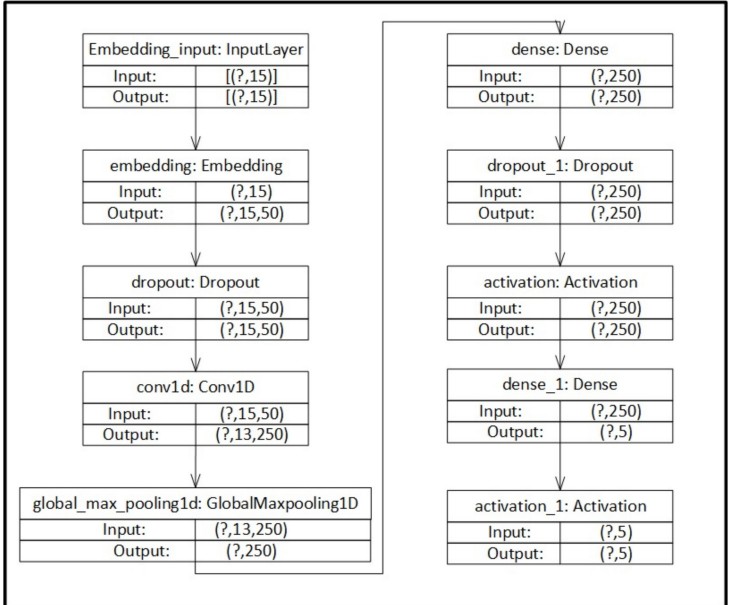

**Figure 2.** Structure of the convolutional neural network (CNN) model used in the experiment. This shows the layers used in the CNN algorithm and their arrangement.

The role of dropout layers is to simplify computational complexity and the convergence of the learning process.

## 4. Results

The results obtained from the multiclass classification can be seen in Table 3.

**Table 3.** Table metrics.

| Algorithm | Metrics | | | | |
|---|---|---|---|---|---|
| | AUC DDOS | AUC DOS | AUC Reconnaissance | AuC Normal | AUC Theft |
| Random Forests (RF) | 1 | 1 | 0.98 | 1 | 0.96 |
| CNN | 0.98 | 0.98 | 0.99 | 0.99 | 1 |
| MLP | 0.56 | 0.51 | 0.97 | 0.99 | 0.99 |

A receiver operating characteristic (ROC) curve shows the performance measurement tool for classification problems in determining the threshold of a model. This ROC curve had two parameters: true positive rate and false positive rate.

An area under the curve (AUC) is used for classification analysis in terms of choosing which model best predicts a class. One example application is the ROC curve. Here, the true positive rate is compared to the false positive rate.

As seen in Figure 3, the RF was good for multiclass classification as a whole.

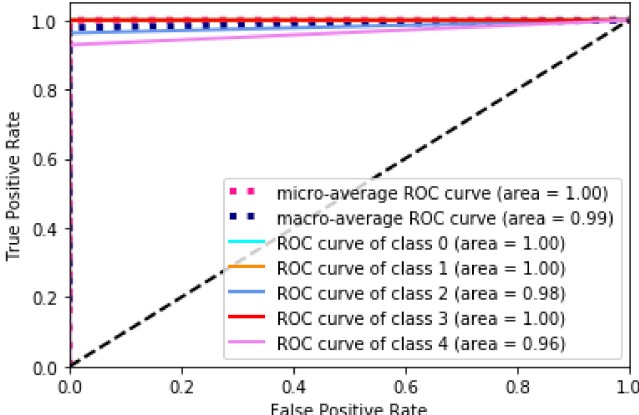

**Figure 3.** Receiver operating characteristic (ROC) curve of the random forest (RF) algorithm.

As seen in Figure 4, the CNN was as good as RF at multiclass classification.

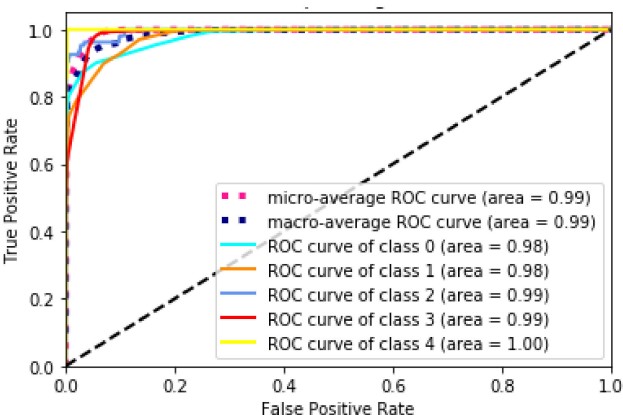

**Figure 4.** ROC curve of the CNN algorithm.

As can be seen in Figure 5, MLP was not as good as RF or the CNN at multiclass classification.

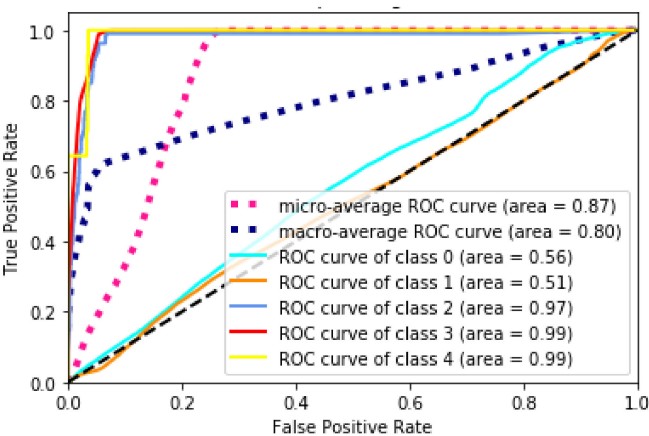

**Figure 5.** ROC curve of the multi-layer perceptron (MLP) algorithm.

From Table 4, with batch size 32, it appears that the mean accuracy increased with increasing number of research epochs for the MLP classifier. For the CNN, there was a slight decrease when the number of epochs increased from 10 to 50.

**Table 4.** Metrics batch size 32.

| Algorithm | Epoch | Mean Accuracy | Elapsed Time |
|---|---|---|---|
| CNN | 10 | 90.85% | 59 min 38 s |
| MLP | 10 | 53.07% | 37 min 8 s |
| CNN | 30 | 89.82% | 155 min 29 s |
| MLP | 30 | 62.95% | 122 min 33 s |
| CNN | 50 | 88.30% | 227 min 21 s |
| MLP | 50 | 62% | 184 min 45 s |

Table 5 shows the result with batch size 64. Here, the mean accuracy decreased with the increasing number of research epochs for the MLP classifier. For the CNN, there was a slight decrease when the number of epochs increased from 10 to 50.

**Table 5.** Metrics batch size 64.

| Algorithm | Epoch | Mean Accuracy | Elapsed Time |
|---|---|---|---|
| CNN | 10 | 91.15% | 20 min 57 s |
| MLP | 10 | 76.92% | 26 min 56 s |
| CNN | 30 | 91.02% | 64 min 18 s |
| MLP | 30 | 54.04% | 64 min 19 s |
| CNN | 50 | 90.64% | 112 min 55 s |
| MLP | 50 | 53.89% | 102 min 20 s |

Table 6 shows the result with batch size 128. It appears that the mean accuracy increased with the increasing number of research epochs for the MLP classifier. For the CNN, there was a slight decrease when the number of epochs increased from 10 to 30, and then it increased at 50 epochs. From Tables 3–5, we can see that the increase in batch size could reduce duration time.

**Table 6.** Metrics batch size 128.

| Algorithm | Epoch | Mean Accuracy | Elapsed Time |
|---|---|---|---|
| CNN | 10 | 90.87% | 11 min 33 s |
| MLP | 10 | 54.10% | 10 min 16 s |
| CNN | 30 | 90.76% | 45 min 44 s |
| MLP | 30 | 54.43% | 27 min 58 s |
| CNN | 50 | 91.27% | 54 min 27 s |
| MLP | 50 | 79.01% | 46 min 18 s |

## 5. Conclusions

In this study, we examined different machine-learning and deep-learning algorithms in an IoT network. We incorporated the evaluation of RF, CNN, and MLP algorithms. Random forests and the CNN provided the best result in terms of accuracy and the AUC for multiclass classification. The addition of epochs in experiments with 32 and 64 batches resulted in a slight decrease in accuracy, whereas in trials with 128 batches, there was a slight increase in accuracy.

We also found that increasing the batch size could speed up the calculation process. Doubling the change in batch size on the MLP could make the calculation process 1.4–2.6 times faster, whereas the CNN could make the calculation process 1.8–2.4 times faster.

In the future, we aim to develop models with different algorithms and also to combine several algorithms for machine learning or deep learning. In addition, this algorithm is expected to be implemented into the NIDS so that it can be used in real time to mitigate attacks.

**Author Contributions:** Conceptualization, B.S. and R.F.S.; methodology, B.S.; software, B.S.; validation, B.S. and R.F.S.; formal analysis, B.S.; resources, B.S. and R.F.S.; data curation, B.S.; writing—original draft preparation, B.S.;

writing—review and editing, R.F.S.; visualization, B.S.; supervision, R.F.S. All authors have read and agreed to the published version of the manuscript.

**Funding:** This research received partial funding from University of Indonesia under PUTI Q2 Grant number NKB-1732/UN2.RST/HKP.05.00/2020.

**Acknowledgments:** We thank the University of Indonesia for financial support for this research under the PUTI Q2 Grant number NKB-1723/UN2.RST/HKP.05.00/2020. The authors would like to express their deep gratitude to the reviewers for their valuable suggestions and important comments that have greatly helped to improve the presentation of this manuscript.

**Conflicts of Interest:** The authors declare no conflict of interest.

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
