# Peer review of "Intrusion Detection in IoT Networks Using Deep Learning Algorithm"

_information, doi:10.3390/info11050279_

Round 1
Reviewer 1 Report
Good work,
Small attention to details is needed. Here is what I spotted while reading the text on page 2 about differences in how you use dots to end sentences: ‘needed [6-11].’ followed by ‘strategy. [12, 13]’
On page 3: you state: ‘Appropriated DoS (DDoS)’, wouldn’t be better to use distributed denial of service?
You need to make sure all abbreviations include full text the first time the abbreviation is introduced, here are some I spotted: BoT-IoT, PCAP, CSV, DT, etc. I know it seems repetative, but many readers would have no idea what you mean unless you include explanations on each abbreviation.
In the conclusion, I found the text describing contributions a bit short. You did a great deal of work, but you are not making it easy for readers to identify what you achieved. A few more words could help researchers identify what your paper is about. Although the current text as it stands it describes the output, you could be a bit more generous with words in the conclusion. Maybe a few words building upon the text I copied would resolve this: ‘Random Forest and Convolutional Neural Network provide the best result in term of accuracy and Area Under Curve for multiclass classification. We also find that adding batch size can speed up the calculation process.’
You could make your contributions more visible in the article by adding a few words to express opinion on how your work compares, or builds upon existing research studies. For example, your article covers intrusion detection in IoT systems and since page 1, you mention ‘Mirai’. Your discussion on previous work on the topic of cyber risk in IoT systems that discussed Mirai is not well covered in your literature review. you should check the most recent literature on this topic, e.g.: https://doi.org/10.1016/j.compind.2018.08.002 and standardisation of IoT risk assessment e.g.: https://doi.org/10.1007/s42452-019-1931-0. In addition, you don’t mention a definition of IoT and the values of IoT, to understand the risk, you need to understand the potential loss (values), e.g.: https://doi.org/10.1057/s41265-018-0054-1
I hope this feedback will help your article gain more attention in the areas you researched.
Reviewer 2 Report
I like the paper. The work is interesting.
Of course, I can see (based on the extension as well as some of the conclusions reached) that it is an ongoing work. I provide some suggestions before the manuscript can be considered for publication in this journal.
- The introduction is short, the authors need to improve it, to extend it, to indicates the main objective of the work
- please, provide more information about the dataset, i.e. a description table
- explain better the figure 2 as it is (at least what I understand) a key part of the work.
- please the quality of the figures needs to be improved, hard to read now (from figure 2)
- The conclusions need to be much more developed.
Above all, an on work project need to detail much more not only the objectives achieved but especially the future lines
Author Response
Please see the attachment.
Thank you for your time.

Round 2
Reviewer 2 Report
I see that the authors have changed most of my suggested changes.
Of course, I still see a little room for improvement, such as to improve image quality.
Author Response
Authors say thank you to the reviewer that has given valuable comment for the article.
Point 1: I still see a little room for improvement, such as to improve image quality.
Response 1: The author has re-draw Figure1 in Line 195 and Figure2 in Line 200.
Thank you for your time.
Best Regards,
Bambang Susilo